# CD30 Lateral Flow and Enzyme-Linked Immunosorbent Assays for Detection of BIA-ALCL: A Pilot Study

**DOI:** 10.3390/cancers15215128

**Published:** 2023-10-25

**Authors:** Victoria G. Zeyl, Haiying Xu, Imran Khan, Jason T. Machan, Mark W. Clemens, Honghua Hu, Anand Deva, Caroline Glicksman, Patricia McGuire, William P. Adams, David Sieber, Mithun Sinha, Marshall E. Kadin

**Affiliations:** 1Division of Plastic Surgery, Department of Surgery, Brown Alpert School of Medicine, Providence, RI 02903, USA; torizeyl@gmail.com; 2Department of Pathology and Laboratory Medicine, Brown Alpert School of Medicine, Providence, RI 02903, USA; haiyingxu1@gmail.com; 3Division of Plastic Surgery, Department of Surgery, Indiana University School of Medicine, Indianapolis, IN 46202, USA; imrkmoha@iupui.edu (I.K.); mitsinha@iu.edu (M.S.); 4Lifespan Biostatistics, Epidemiology, Research Design, and Informatics (BERDI) Lifespan Hospital System, Providence, RI 02903, USA; jmachan@lifespan.org; 5Division of Plastic and Reconstructive Surgery, MD Anderson Cancer Center, University of Texas, Houston, TX 77030, USA; mwclemens@mdanderson.org; 6Macquarie Medical School, Macquarie University, Sydney, NSW 2109, Australia; helen.hu@mq.edu.au (H.H.); anand.deva@ishc.org.au (A.D.); 7Plastic & Reconstructive Surgery, Faculty of Health and Medical Science, Macquarie University, Sydney, NSW 2109, Australia; 8Glicksman Plastic Surgery, Sea Girt, NJ 08750, USA; docglicksman@aol.com; 9Parkcrest Plastic Surgery, St. Louis, MO 63141, USA; patricia.mcguiremd@gmail.com; 10Department of Plastic Surgery, University of Texas Southwestern, Dallas, TX 75390, USA; wpajrmd@dr-adams.com; 11Sieber Plastic Surgery, San Francisco, CA 94108, USA; siedav@gmail.com

**Keywords:** CD30, lateral flow assay, lymphoma, diagnosis, screening, ELISA

## Abstract

**Simple Summary:**

A rare complication of breast implants is late development of a lymphoma in fluid accumulating around the implant commonly presenting as unexplained swelling of the breast. This lymphoma is usually curable by removal of the implant and surrounding capsule, but if not detected early can spread to adjacent tissues and lymph nodes requiring radiation, chemotherapy, or immunotherapy. Current diagnosis of the lymphoma requires several time-consuming and costly methods of laboratory testing of 10 or more milliliters of fluid by pathologists. This research describes a method to detect the lymphoma rapidly using only 1 milliliter of fluid, similar to testing for COVID-19.

**Abstract:**

Introduction: Breast Implant-Associated Anaplastic Large Cell Lymphoma (BIA-ALCL) commonly presents as a peri-implant effusion (seroma). CD30 (TNFRSF8) is a consistent marker of tumor cells but also can be expressed by activated lymphocytes in benign seromas. Diagnosis of BIA-ALCL currently includes cytology and detection of CD30 by immunohistochemistry or flow cytometry, but these studies require specialized equipment and pathologists’ interpretation. We hypothesized that a CD30 lateral flow assay (LFA) could provide a less costly rapid test for soluble CD30 that eventually could be used by non-specialized personnel for point-of-care diagnosis of BIA-ALCL. Methods: We performed LFA for CD30 and enzyme-linked immunosorbent assay (ELISA) for 15 patients with pathologically confirmed BIA-ALCL and 10 patients with benign seromas. To determine the dynamic range of CD30 detection by LFA, we added recombinant CD30 protein to universal buffer at seven different concentrations ranging from 125 pg/mL to 10,000 pg/mL. We then performed LFA for CD30 on cryopreserved seromas of 10 patients with pathologically confirmed BIA-ALCL and 10 patients with benign seromas. Results: Recombinant CD30 protein added to universal buffer produced a distinct test line at concentrations higher than 1000 pg/mL and faint test lines at 250–500 pg/mL. LFA produced a positive test line for all BIA-ALCL seromas undiluted and for 8 of 10 malignant seromas at 1:10 dilution, whereas 3 of 10 benign seromas were positive undiluted but all were negative at 1:10 dilution. Undiluted CD30 LFA had a sensitivity of 100.00%, specificity of 70.00%, positive predictive value of 76.92%, and negative predictive value of 100.00% for BIA-ALCL. When specimens were diluted 1:10, sensitivity was reduced to 80.00% but specificity and positive predictive values increased to 100.00%, while negative predictive value was reduced to 88.33%. When measured by ELISA, CD30 was below 1200 pg/mL in each of six benign seromas, whereas seven BIA-ALCL seromas contained CD30 levels > 2300 pg/mL, in all but one case calculated from dilutions of 1:10 or 1:50. Conclusions: BIA-ALCL seromas can be distinguished from benign seromas by CD30 ELISA and LFA, but LFA requires less time (<20 min) and can be performed without special equipment by non-specialized personnel, suggesting future point-of-care testing for BIA-ALCL may be feasible.

## 1. Introduction

Breast Implant-Associated Anaplastic Large Cell Lymphoma (BIA-ALCL) has been recently recognized by the World Health Organization as a T-cell lymphoma associated with breast implants in women after reconstructive surgery for breast cancer, prophylactic mastectomy because of high genetic risk for breast cancer, or other cosmetic reasons [1]. At present, there are over 35 million women living with breast implants who have an estimated 1 in 3000–7000 risk of developing BIA-ALCL [2]. In one prospective cohort study of patients with textured implants, the overall risk of BIA-ALCL was 1/355 women or 0.311 cases per 1000 person-years (95% CI 0.118 to 0.503) [3].

About 80% of patients present with breast swelling, pain, and/or erythema associated with a unilateral effusion (seroma) confined to the capsule of a textured surface breast implant a median time of 9 years after the initial implant has been placed [4]. The remaining patients develop infiltrative disease forming tumors in breast parenchyma, the chest wall, and/or regional lymph nodes, requiring radiation, chemotherapy, or immunotherapy [5]. Patient survival is significantly improved by detection of BIA-ALCL when remaining localized to seromas or the lining of the peri-implant capsule [6]. Thus, it is important to detect disease in these early stages.

Current best practice guidelines for pathologic diagnosis of BIA-ALCL are to produce air-dried smears stained with Wright–Giemsa stain, a cell block for hematoxylin and eosin staining, detection of CD30 by immunohistochemistry and polymerase chain reaction, and T-cell receptor gene rearrangement to detect clonality [7]. Using flow cytometry to detect CD30 and other cell markers is another approach [8,9]. All have in common high cost and delay of diagnosis. In a landmark paper, Hanson et al. demonstrated CD30 enzyme-linked immunosorbent assay (ELISA) as an alternative to CD30 immunohistochemistry for the screening of patient seromas for BIA-ALCL [10]. However, ELISA requires multiple reagents, specially prepared plates, and an ELISA plate reader, and is impractical for point-of-care diagnosis. Here we build on the ELISA approach to introduce a lateral flow assay (LFA) that can be delivered at point-of-care for detection of CD30 in BIA-ALCL effusions/seromas within 20 min by unspecialized personnel using less than 1 mL of fluid. LFA has been used for rapid detection of COVID-19 [11] and intraoperatively to detect IL-6 in synovial fluid of peri-prosthetic infections [12], demonstrating the utility of this approach.

## 2. Methods

### 2.1. Patients

We evaluated 25 cryopreserved peri-implant seromas collected from patients at Rhode Island Hospital and MD Anderson Hospitals in the USA, and Macquarie University Medical School in Sydney, Australia. Informed consent and institutional approvals were obtained. All patients with seroma samples were Caucasian women who had textured breast implants. Based on morphological and immunohistochemical analyses, seromas were diagnosed as malignant (15 BIA-ALCL samples) or benign (10 samples).

### 2.2. Specimens

Seromas and were centrifuged, and supernatants stored at −80 °C. Cell pellets were used for cytology and cell block preparation for histology and immunohistochemistry.

### 2.3. Pathology and Immunohistochemistry

To determine the cellular content of seromas, cytospins were performed with a Shandon Scientific 74000102 CytoSpin 3 Cytocentrifuge at 1000 rpm for three min. Following cold acetone fixation and air drying, slides were stained with Wright–Giemsa stain and with Ber-H2 antibody against CD30 to confirm the presence of tumor cells using 1:40 concentration of Ber-H2 anti-CD30 mouse IgG1-kappa monoclonal antibody (Sigma-Aldrich, St. Louis MO, USA) and an anti-mouse IgG detection kit from R&D Systems, Minneapolis, MN, USA.

### 2.4. Lateral Flow Assay (LFA)

The lateral flow assay was performed using Universal LFA strips from Abcam (Waltham MA, USA), which consist of a nitrocellulose matrix membrane with pores between 0.2 and 2 um in size, containing a test line of immobilized anti-Ulfa-Tag antibody that binds an Ulfa-Tag conjugated capture antibody, which further binds CD30 in complex with a gold-detection antibody. Undiluted or specified dilutions of seroma fluid were placed into wells with CD30 capture antibody (clone Monoclonal, EPR21828-36/cat# ab244785, Abcam), CD30 detection antibody (Recombinant Monoclonal, RM425/Cat# MA5-36219, Thermo Fisher Scientific, Waltham MA, USA), and gold biotin. A red-colored control (C-line) was produced by immobilized streptavidin and gold nanoparticles conjugated to biotin. The sample test strip was then placed into each well and monitored for twenty minutes. A positive result was a red test line whose intensity corresponded to the concentration of CD30. The control line ensured the flow of the antibodies and confirmed a negative result. Duplicate strips were used for each test. A detailed LFA protocol is shown in Appendix A. To estimate the concentration of CD30 in clinical specimens, recombinant CD30 protein (Abcam, ab140584) was added to universal buffer and the intensity of the red test line compared to a scale provided by Abcam.

### 2.5. Enzyme Linked Immunosorbent Assay (ELISA)

CD30 ELISA was performed using Human CD/TNFRSF8 DuoSet ELISA kit DY6126-05 and a DuoSet Ancillary Reagent Kit 2 DY008 from R&D Systems, Minneapolis, MN, USA) on samples that were undiluted, and diluted 1:10 and 1:50. Clinical samples were tested at 1:10 and 1:50 because lower dilutions were found to be unreliable by Hanson et al. [10]. CD30 capture antibody was used to coat the bottom of wells. After washing three times, standards and clinical samples were dispersed into flat-bottom wells and incubated at 4 °C overnight. Wells were washed three times again and a detection CD30 antibody was added for two hours; the wells were washed again three times with buffer, and then color reagents A and B added successively followed by stop solution, before reading at 450 nanometers on a chromogenic reader, model number 550, from Bio-Rad Laboratories, Hercules, CA, USA.

## 3. Results

### 3.1. Pathology

The diagnosis of BIA-ALCL or benign seroma was submitted by the local pathologist of record who examined the original seroma fluid. This information was provided to us in the pathology report with patient identifiers deleted. Cytospin cytology of fresh seroma from BIA-ALCL patient 61 revealed a monotonous population of large CD30+ cells, whereas seroma of BIA-ALCL patient 62 had few CD30+ cells surrounded by erythrocytes and CD30 negative small lymphocytes (Figure 1).

### 3.2. Lateral Flow Assay

Recombinant CD30 protein added to universal buffer demonstrated that sCD30 produced a distinct test line at concentrations higher than 2000 pg/mL and faint test lines at 250–500 pg/mL. (Figure 2). As a result, 3 of 10 benign seromas yielded faint test lines in undiluted seromas but were negative at 1:10 dilution, whereas all undiluted BIA-ALCL seromas were strongly positive and 8 of 10 BIA-ALCL retained a positive test line at 1:10 dilution (Figure 3). Undiluted CD30 LFA had a sensitivity of 100.00%, specificity of 70.00%, positive predictive value of 76.92%, and negative predictive value of 100.00% for BIA-ALCL (Table 1). When specimens were diluted 1:10, sensitivity was reduced to 80.00% but specificity and positive predictive values increased to 100.00%, while negative predictive value was reduced to 88.33%. (Table 2). Percentages for sensitivity and specificity are presented with their 95% confidence intervals (Wilson Score). Note that a clear positive result was present in two bloody BIA-ALCL seromas (M2 and M7).

### 3.3. ELISA

ELISA results on cryopreserved seromas showed all benign seromas had undiluted CD30 concentrations below 1200 pg/mL (Figure 4). Four of six benign seromas contained >500 pg/mL CD30 and five of six contained >250 pgL. In contrast, in all but one case (1817), BIA-ALCL seromas contained CD30 levels > 2300 pg/mL, calculated from dilutions of 1:10 or I:50. Seroma 61 with many CD30+ large cells had approximately five times higher CD30 concentration than seroma 62 with fewer CD30+ cells (Figure 4). A 10-fold dilution of seroma 61 resulted in an expected decrease in CD30 concentration from 7526 pg/mL to 708 pg/mL, which was visible as a positive test line (Figure 3).

## 4. Discussion

This study demonstrates the feasibility of CD30 LFA for future point-of-care diagnosis of BIA-ALCL. Current requirements for diagnosis of BIA-ALCL include demonstration of atypical/anaplastic cells that express CD30 by immunohistochemistry or flow cytometry [13]. These tests require 10–50 mL of fluid, specialized equipment, trained personnel, e.g., pathologists, delay of diagnosis, multiple patient visits to multiple facilities, and increased costs. Our experience is that CD30 LFA requires less than 1 mL of fluid and no more than 20 min to interpret. CD30 LFA during standard of care initial and follow-up visits or intra-operative evaluation of a peri-prosthetic seroma will provide physicians immediate feedback for patient management. A positive test will prompt ordering of additional breast evaluation with magnetic resonance imaging if not yet obtained, positron emission tomography to further delineate any local or distant disease, and potentially alter the surgical plan if malignancy is suspected. Because breast implants are used in rural USA, and under-developed regions of South America and Asia, the LFA could make affordable point-of-care testing available to underserved patients.

Due to the rarity of the disease, our study was limited to a small number of malignant seromas and it was not possible to correlate LFA with ELISA results for most BIA-ALCL patients. A positive LFA does not determine if the lymphoma is localized to the fluid or has infiltrated the capsule. However, a positive LFA should alert the clinician of the need to do further staging. LFA was performed on cryopreserved specimens and it is likely that more reliable results will be obtained from fresh specimens obtained at point-of-care in a planned prospective study. The CD30 LFA is very sensitive and proper dilution of seromas is critical. We observed that 1:10 dilution works best both for benign and malignant seromas for CD30 detection. In undiluted benign seromas from 3 of 10 patients, we observed a faint coloration at the T-line, which was not detected at 1:10 dilution, whereas a positive test line was observed in 8 of 12 BIA-ALCL seromas diluted 1:10. It remains to be determined if CD30 LFA is positive in peri-implant seromas due to EBV+ large B-cell lymphoma in which CD30+ cells can be found [14]. Immunohistochemistry positive for B-cell markers, such as CD20, CD79a, or PAX5, and EBV-encoded small RNA (EBER), will clarify the differential diagnosis. The CD30 LFA is expected to be negative in squamous cell carcinomas recently reported around breast implants [15,16]. Interestingly, one malignant seroma was originally classified as benign, but was re-evaluated and reclassified as ALCL when it was found to have a high level of IL-9 and a positive IL-10 LFA [17]. Another patient with BIA-ALCL was originally diagnosed by an outside referral laboratory as having recurrent breast carcinoma. This patient had a positive LFA and upon referral to another medical center was confirmed to have BIA-ALCL.

Hanson et al. reported that CD30 was not detected by ELISA in any of seven benign seromas. However, the mean concentration of CD30 was higher in benign seromas contralateral to breast implants affected by BIA-ALCL than in other benign seromas [10]. We consistently detected CD30 as a faint positive test line in LFA of 3 of 10 clinically benign seromas and by ELISA in four of six benign seromas consistent with detection of recombinant CD30 protein at 500 pg/mL in dilution experiments. Detection of CD30 in benign seromas raises the hypothesis that CD30 activation could be a marker of early or premalignant disease. Supporting this hypothesis, Di Napoli reported oligoclonal expansion of T cells in a lymphocyte-rich benign seroma with 5% CD30+ cells [18]. We also reported clustered CD30+ cells with the morphology of ALCL surrounded by inflammatory cells in a capsule contralateral to a breast with BIA-ALCL [19]. The CD30+ cells in the benign contralateral capsule lacked the tumor marker pSTAT3 [20,21] found in the BIA-ALCL breast capsule.

One of the most compelling advantages of using a CD30 lateral flow assay (LFA) over the traditional enzyme-linked immunosorbent assay (ELISA) or immunohistochemistry for detecting BIA-ALCL lies in its cost-efficiency. At approximately USD 0.10–USD 3.00 per test, the LFA method represents a significant cost reduction compared to approximately USD 14 per result for ELISA, based on a 96-well plate costing around USD 550. Even more striking is the difference in cost between LFA and traditional CD30 immunohistochemistry, which stands at approximately USD 250 at least per test, with some hospitals fee schedules in excess of USD 1000 per test. The potential cost savings for both private practice and hospitals are substantial when transitioning from either ELISA or traditional immunohistochemistry to LFA. For a private practice performing two tests per week, savings could range from USD 1300 to USD 25,844 per year. For a hospital conducting 50 tests per week, the savings could be even more dramatic, ranging from USD 32,500 to USD 646,100 annually. This vast cost difference could lead to more widespread testing and could potentially make screening for BIA-ALCL more accessible to lower-income populations, developing countries, or healthcare systems with limited resources.

Another crucial benefit of the CD30 LFA is its speed and user friendliness. As established in the study, results can be rapidly acquired in less than 20 min. This is significantly faster compared to immunohistochemistry, which usually takes 7–10 days, or even ELISA, which, despite being relatively quick, still requires specialized equipment and preparation time. The study showed that the CD30 LFA has high diagnostic sensitivity and specificity, particularly when the specimens are diluted 1:10. While the sensitivity dropped from 100% to 80% upon dilution, specificity and positive predictive values increased to 100%. This represents a promising diagnostic tool with minimal risk of false-negative results. It does pose some challenges related to false-positive results (especially undiluted), but this could potentially be mitigated by further studies optimizing the assay conditions or pairing it with other diagnostic methods as part of a diagnostic panel. Although LFA showed high sensitivity and specificity, it is essential to consider that this was a small-scale pilot study. Larger studies are necessary to confirm these findings. Moreover, while the assay is simple enough to be performed by non-specialized personnel, training would still be required to ensure accurate and reproducible results. Given that BIA-ALCL can present similarly to benign seromas, clinical correlation and possibly supplemental testing would still be needed to definitively confirm a diagnosis. Presently, the LFA is best used as a screening test to prompt further evaluation with pathologic and radiologic studies.

The CD30 lateral flow assay offers a significantly less expensive, quicker, and easier-to-use alternative to both ELISA and traditional immunohistochemistry for the detection of BIA-ALCL. Its potential for point-of-care application could revolutionize early detection and management of this malignancy, making testing more accessible and timelier. While additional large-scale studies are needed to validate these preliminary findings, the initial results are promising and suggest that LFA could be an essential tool in the evolving diagnostic landscape for BIA-ALCL.

## 5. Conclusions

We conclude that CD30 LFA has potential for rapid detection of BIA-ALCL in patients with delayed seromas. The test is portable, low-cost and can be performed by surgeons in consultation with breast implant patients in an office or surgical suite. Further, non-specialized personnel can perform the CD30 LFA, making it feasible for use in underserved populations. Patients will be spared added time away from their family and added travel expenses, while their physicians can plan the evaluation of the disease stage necessary for definitive treatment by local surgery or adjuvant therapies.

## 6. Patents

Rhode Island Hospital and Indiana University have a provisional patent application (International application # PCT/US2023/012086) on behalf of Imran Khan, Mithun Sinha and Marshall E Kadin relating to the LFA based diagnosis of BIA-ALCL.

## Figures and Tables

**Figure 1 cancers-15-05128-f001:**
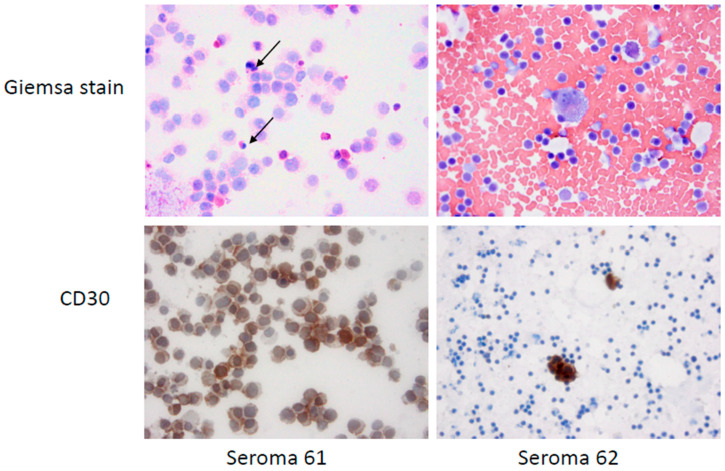
Cytospin cytology of fresh seromas from two patients with pathologically diagnosed BIA-ALCL. Giemsa stain of seroma 61 shows monotonous population of anaplastic cells which are CD30+. Apoptotic bodies of dead tumor cells are seen (arrows). Seroma 62 contains a CD30+ large multinucleate tumor cell and a CD30+ large mononuclear tumor cell surrounded by small CD30 negative lymphocytes and many erythrocytes (600×).

**Figure 2 cancers-15-05128-f002:**
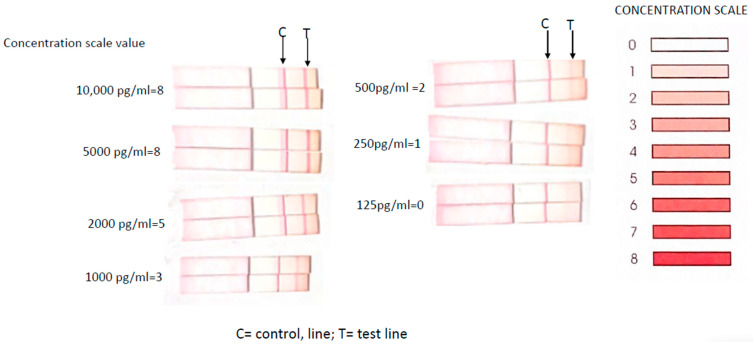
LFA of recombinant CD30 protein at different concentrations produces varied intensity test lines corresponding to different intensities on the color scale. CD30 could be detected at concentrations as low as 250 pg/mL.

**Figure 3 cancers-15-05128-f003:**
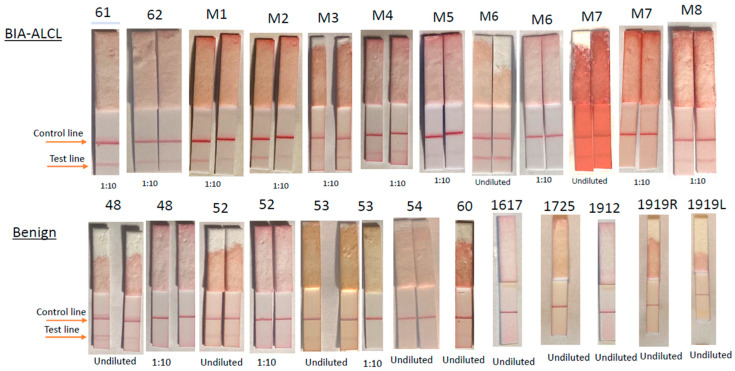
LFA of clinical specimens produces positive test line in 3 of 10 undiluted benign seromas (48,52,53) eliminated at 1:10 dilution, while all but 2 (M6, M7) of 10 malignant seromas remain positive at 1:10 dilution.

**Figure 4 cancers-15-05128-f004:**
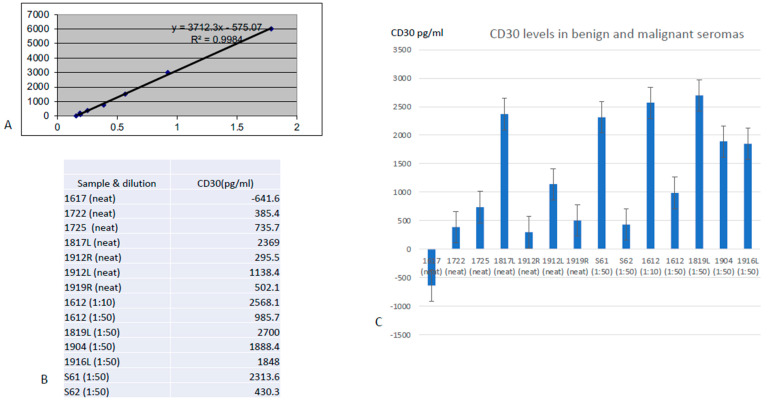
Representative ELISAs of benign and malignant seromas. (**A**) Straight line values using manufacturer’s standards. (**B**) Table of numerical values determined for patient samples. The optical density of sample 1617 is apparently lower than what the lowest standard captured. (**C**) Bar graph showing CD30 levels below 1200 pg/mL in six undiluted benign seromas, while undiluted BIA-ALCL seroma 1817L contained 2369 pg/mL CD30 and six other BIA-ALCL seromas require dilutions of 1:10 or 1:50 to not exceed 2300 pg/mL. Standard error bars are shown. 1912L and 1912R refer to left and right seromas of the same patient.

**Table 1 cancers-15-05128-t001:** Statistical analysis of CD30 LFA in undiluted seromas.

CD30 LFA	#	##	p	LCL	UCL	SE	z
Sensitivity	10	10	100.00%	65.55%	100.00%	0	1.9
Specificity	7	10	70.00%	37.37%	91.91%	0.14	1.9
Positive Predictive Value	10	13	76.92%	45.98%	93.84%	0.12	1.9
Negative Predictive Value	7	7	100.00%	56.09%	100.00%	0	1.9

# is numerator; ## is denominator. Sensitivity defined as the % malignant who the assay defined as positive. Specificity defined as the % benign who the assay defined as negative. Positive Predictive Value defined as the % the assay defined as positive who turned out to be malignant. Negative Predictive Value defined as the % the assay defined as negative who turned out to be benign.

**Table 2 cancers-15-05128-t002:** Statistical analysis of CD30 LFA in seromas diluted 1:10.

CD30 LFA 1:10	#	##	p	LCL	UCL	SE	z
Sensitivity	8	10	80.00%	44.22%	96.46%	0.13	1.9
Specificity	10	10	100.00%	65.55%	100.00%	0	1.9
Positive Predictive Value	8	8	100.00%	59.77%	100.00%	0	1.9
Negative Predictive Value	10	12	83.33%	50.88%	97.06%	0.11	1.9

# is numerator; ## is denominator. Sensitivity defined as the % malignant who the assay defined as positive. Specificity defined as the % benign who the assay defined as negative. Positive Predictive Value defined as the % the assay defined as positive who turned out to be malignant. Negative Predictive Value defined as the % the assay defined as negative who turned out to be benign.

## Data Availability

All data has been included in the manuscript.

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
