# Peer review of "CD30 Lateral Flow and Enzyme-Linked Immunosorbent Assays for Detection of BIA-ALCL: A Pilot Study"

_cancers, 2023, doi:10.3390/cancers15215128_

Round 1
Reviewer 1 Report
The authors report a lateral flow assay (LFA) and enzyme-linked immunosorbent assay (ELISA) for CD30 detection in seromas.
General:
Although these assays may be helpful to triage or confirm the presence of CD30. Morphology is essential to diagnosis a malignant lymphoma in breast implants.
These assays cannot determine if anaplastic large cell lymphoma (ALCL) is located only within the fluid or if it has infiltrated the capsule.
Comment:
Unfortunately, LFA may detect CD30 expression but pathology is necessary to confirm the diagnosis of ALCL as CD30 expression can be seen in other entities benign and malignant. LFA also is somewhat finicky.
Proper dilution of the seroma fluid is also necessary to get an accurate result from LFA and these tumors are rare making the skill of the user less reliable.
As an ancillary test, LFA may be useful but it would not be adequate as a diagnostic test.
Author Response
We thank the reviewers for their timely and thoughtful critiques. We have addressed all the comments and modified the text accordingly using track changes. We also thought to add to the title “A Pilot Study” as this is a preliminary study we hope to continue if funding becomes available, which publication of this study should help to obtain. The following lists reviewer comments and our responses.
The authors report a lateral flow assay (LFA) and enzyme-linked immunosorbent assay (ELISA) for CD30 detection in seromas.
General:
Comment: Although these assays may be helpful to triage or confirm the presence of CD30. Morphology is essential to diagnosis a malignant lymphoma in breast implants.
Response: We agree the LFA is best used to triage or confirm the presence of CD30 but morphology remains essential to confirm the diagnosis of BIA-ALCL. We have added that “Presently the LFA is best used as a screening test to prompt further evaluation with pathologic and radiologic studies.” Lines 276-8
However, the rapid point-of-care LFA testing could save patient time away from family, travel and other expenses. The LFA test could be especially useful in screening patients in rural and underserved areas
Comment: These assays cannot determine if anaplastic large cell lymphoma (ALCL) is located only within the fluid or if it has infiltrated the capsule.
Response: We agree that CD30 in seroma fluid does not discount spread of tumor to the capsule or beyond. A sentence has been added to the discussion to make this point. Lines 213-16. The focus of this study is to come up with an early diagnosis of BIA-ALCL and test its validity and sensitivity on seromas of patients. The study is not intended to determine the stage or disease progression.
Comment: Unfortunately, LFA may detect CD30 expression but pathology is necessary to confirm the diagnosis of ALCL as CD30 expression can be seen in other entities benign and malignant. LFA also is somewhat finicky.
Response: We agree that CD30 in seroma fluid is not specific and pathology is necessary for the diagnosis of BIA-ALCL. However, when specimens were diluted 1:10, specificity and positive predictive values increased to 100.00%, potentially alerting clinicians that additional testing including cytology, histology and ancillary testing e.g. flow cytometry or immunohistochemistry is essential.
Proper dilution of the seroma fluid is also necessary to get an accurate result from LFA and these tumors are rare making the skill of the user less reliable.
Response: Dilution of samples is easily accomplished and the eventual test kit will be constructed to have a ready-made dilution capacity. When specimens were diluted 1:10, specificity and positive predictive values increased to 100.00%, potentially alerting clinicians that additional testing including cytology, histology and ancillary testing e.g. flow cytometry or immunohistochemistry is essential
As an ancillary test, LFA may be useful but it would not be adequate as a diagnostic test.
Response: Agree. However, the ability to test a seroma in the physician’s office will help communicate the need for further testing to the patient. Additional fluid is routinely sent off for full diagnostic testing and a positive LFA suggests the need for scheduling of radiologic imaging.
Reviewer 2 Report
The authors have developed a method to detect free CD30 protein in seromas using LFA as an alternative to the costly and time-consuming ELISA for the diagnosis of Breast Implant-Associated Anaplastic Large Cell Lymphoma (BIA-ALCL), which is rare in patients with breast implants.
Major comments
This study has poor scientific credibility because the results of conventional diagnostic methods, such as pathology and flow cytometry, are not fully presented, and it is not possible to confirm whether the diagnosis of the specimens used in this study was correct.
Although considered as an alternative method to ELISA, few lymphoma case samples have been examined using ELISA, and the correlation between LFA and ELISA data is not known. In this study, preserved specimens were used, but unless similar results can be reproduced with fresh specimens, it is not clear whether the ELISA is clinically useful. Further studies are needed to determine how the results are affected by the quality of the specimens in a large number of cases.
The positive/negative LFA test is qualitative and depends on the examiner's subjective judgment, and although it is simple, there is concern about misdiagnosis. It is understandable that the diagnosis of BIA-ALCL requires low cost, but rapidity is not necessary.
This study is a unique concept and could be useful as a screening test. Repeated and longitudinal testing may allow for an early diagnosis of BIA-ALCL. I look forward to a report of a prospective study in a larger number of cases.
Minor comments
Chapter 2.4 is missing a subject in the write-up.
Do the L and R in the sample name mean left and right of the same patient? I would like an explanation.
Author Response
We thank the reviewers for their timely and thoughtful critiques. We have addressed all the comments and modified the text accordingly using track changes. We also thought to add to the title “A Pilot Study” as this is a preliminary study we hope to continue if funding becomes available, which publication of this study should help to obtain. The following lists reviewer comments and our responses.
The authors have developed a method to detect free CD30 protein in seromas using LFA as an alternative to the costly and time-consuming ELISA for the diagnosis of Breast Implant-Associated Anaplastic Large Cell Lymphoma (BIA-ALCL), which is rare in patients with breast implants.
Major comments
This study has poor scientific credibility because the results of conventional diagnostic methods, such as pathology and flow cytometry, are not fully presented, and it is not possible to confirm whether the diagnosis of the specimens used in this study was correct.
Response: The diagnosis of BIA-ALCL or benign seroma was submitted by the local pathologist of record who examined the original seroma fluid. This information was provided to us in the pathology report with patient identifiers deleted. We added this information to the methods in the revised version.
Comment: Although considered as an alternative method to ELISA, few lymphoma case samples have been examined using ELISA, and the correlation between LFA and ELISA data is not known. In this study, preserved specimens were used, but unless similar results can be reproduced with fresh specimens, it is not clear whether the ELISA is clinically useful. Further studies are needed to determine how the results are affected by the quality of the specimens in a large number of cases.
Response: We agree that fresh specimens would likely be more reliable making the LFA a clinically useful test. We plan to do a prospective study performing LFA upon receipt. Meanwhile, we hope these preliminary studies will prompt others to test this LFA on fresh specimens, followed by routine pathological exam and flow cytometry of immunohistochemistry. We added a sentence to make this clear in our Discussion.
Comment: The positive/negative LFA test is qualitative and depends on the examiner's subjective judgment, and although it is simple, there is concern about misdiagnosis. It is understandable that the diagnosis of BIA-ALCL requires low cost, but rapidity is not necessary.
Response: We agree that although simple to perform some training for users is essential to avoid misdiagnosis. The color guide and experience with sample dilutions provide guidance to interpretation. Moving forward, we plan to provide samples spiked with known concentrations of recombinant CD30 protein, as show in Figure 2, to train selected users prior to clinical use. With respect to rapidity of diagnosis, plastic surgeons make the decision to operate on a subject often without sampling seroma fluid or seroma fluid is first identified in the operating theater. The ALCL work-up depends on the level of suspicion and a rapid LFA test could help to obtain the proper clinical work-up prior to surgery.
This study is a unique concept and could be useful as a screening test. Repeated and longitudinal testing may allow for an early diagnosis of BIA-ALCL. I look forward to a report of a prospective study in a larger number of cases.
Response: We agree that the LFA may be most useful as a screening test. We hope this preliminary study will enable funding of a prospective testing in a larger number of cases. A sentence has been added to the discussion “Presently the LFA is best used as a screening test to prompt further evaluation with pathologic and radiologic studies.”
Minor comments
Chapter 2.4 is missing a subject in the write-up.
Response: Thank you for mentioning that chapter 2.4 was missing a subject which was a formatting error, now corrected.
Do the L and R in the sample name mean left and right of the same patient? I would like an explanation.
Response: You are correct L and R refer to left and right breast seromas of the same patient. We will make this clear.
As stated in the response to Reviewer 2 and in the Discussion of revised manuscript, it was not possible to correlate LFA with ELISA results for most BIA-ALCL patients. This was due to insufficient sample, a limitation we will avoid in future studies.
Round 2
Reviewer 2 Report
In response to the reviewer's comments, the authors have revised the paper to indicate that this study is a pilot study and that they intend to clarify the scientific validity of the simple and rapid screening method for BIA-ALCL proposed in this paper in a future prospective study. I look forward to the results of this future study.